# Comparative Study of the Nutritional Composition and Antioxidant Ability of Soups Made from Wild and Farmed Snakehead Fish (*Channa Argus*)

**DOI:** 10.3390/foods11203294

**Published:** 2022-10-21

**Authors:** Mengting Ren, Tao Yin, Juan You, Ru Liu, Qilin Huang, Shanbai Xiong

**Affiliations:** 1Key Laboratory of Environment Correlative Dietology (Ministry of Education), College of Food Science and Technology, Huazhong Agricultural University, Wuhan 430070, China; 2National R & D Branch Center for Conventional Freshwater Fish Processing, Wuhan 430070, China

**Keywords:** growth environment, snakehead fish, fish soup, amino acid composition, fatty acid composition

## Abstract

In recent years, fish soup has become an important product for commercial processing of fish due to its health effects. In this study, nutritional composition and antioxidant ability of soups prepared from farmed and wild snakehead fish were analyzed (hereafter FS and WS soup, respectively). For the FS soup, the proximate composition of protein, fat, ash, free amino acids, and soluble peptides were 2.55%, 0.89%, 0.92%, 0.47%, and 0.62%, respectively. The total amino acid was 390.11 mg/ g, and the proportion of essential amino acid was 27.59%. The total fatty acid was 13.64 g/100 g, of which monounsaturated fatty acid was 5.78 g/100 g, n-6 polyunsaturated fatty acid 3.50 g/100 g, and n-3 polyunsaturated fatty acid 0.41 g/100 g, respectively. The contents of Zn and Ca were 9.04 mg/ kg and 1.13 mg/ g, respectively. The DPPH radical-scavenging ability, Fe^2+^ chelating ability, and hydroxyl radical-scavenging ability was 57.89%, 21.21%, and 25.61%, respectively. Overall, there was no obvious difference in the nutritional composition and antioxidant activity between the FS and WS soups. The protein content (1.90%) of the WS soup was relatively lower, but the total fatty acid (16.22 g/100 g), MUFA (7.17 g/100 g), and Zn (12.57 mg/ kg) contents were significantly higher.

## 1. Introduction

Snakehead fish (*Channa argus*), commonly known as black fish, fortune fish, etc., belongs to the family Channidae, and is a ferocious carnivorous freshwater fish [1]. It is widely distributed in rivers, lakes, ponds, and swamps in China, Korea, Japan, Southeast Asia, India, and the Russian Far East [1,2]. The volume of snakehead fish reached 540,000 tons in 2021 in China [3]. Compared to other freshwater fish, snakehead fish has the advantages of high meat yield, high protein and other nutrient content, and various therapeutic effects [4].

Snakehead fish is mainly farmed in high density or grows wildly in fish ponds. The findings of Fuentes, et al. [5] showed differences in proximate composition, color, texture (especially), fatty acids, and free amino acids between the farmed and wild sea bass. O’Neill, et al. [6] did not observe differences in proximate composition between the farmed and wild yellowtail fish. However, from a nutritional point of view, they suggested that the wild yellowtail fish contained better fatty acid complexes and thus might have better nutritional value. Cahu, et al. [7] claimed that the fish farmed under suitable conditions was at least as beneficial in nutrients as wild fish. It can be seen that the influence of growth environment (farm versus wild) on the nutritional composition of fish is controversial, and further research is needed. The major nutritional composition of snakehead fish muscle is water (74.65–79.95%), protein (17.63–20.03%), fat (0.10–1.24%), and ash (1.09–2.16%) [8]. The nutritional composition of snakehead fish muscle may be affected by factors such as origin, food, growth environment, as well as fish size, species, farming time, and processing [6,9,10,11].

In recent years, fish soup has become an important product for commercial processing of fish due to its health effects. The soup prepared from the snakehead fish can promote the healing of wounds and burns [12]. Yuan, et al. [13] reported that snakehead soup had an anti-fatigue effect. In recent years, according to the research results of Zhang, et al. [4], snakehead soup also possessed antioxidant effects. The growth environment may affect the muscle composition of snakehead fish [14], which in turn affects the nutritional composition and functionality of the soup made from the snakehead fish. However, there is no report on the effects of growth environment on the nutritional composition and functionality of snakehead fish soup.

Therefore, the nutritional composition (including proximate composition, amino acid composition, fatty acid composition, and mineral composition) and antioxidant ability of the two soups prepared from the farmed and wild snakehead fish were comparatively analyzed in this study, in order to provide a basis for the nutritional evaluation of the snakehead fish.

## 2. Materials and Methods

### 2.1. Materials and Reagents

In this study, the farmed snakehead fish grew in the fish ponds in Guangzhou. It was farmed alone in high density and fed regularly. The wild snakehead fish of single or very little popularity (<5 tails per pond) grew in the fish ponds in Guangzhou with other farmed fish, and hunted small trash fish, crabs, frogs, etc., for food. The morphology of the two kinds of fish is shown in Appendix A. The comparison of the growth environment and food nutrient composition is shown in Appendix A. The length-to-height ratio and BMI (Body Mass Index) were 4.74 and 5.20, respectively. For the wild snakehead fish, they were 5.81 and 4.24, respectively. Approximate 15 kg raw fish each was taken for the study.

Refined cooking salt (China Salt Industry Group Co., Beijing, China) and peanut oil (Shandong Luhua Group Co., Shandong, China) are commercially available. Copper sulfate, potassium sulfate, concentrated sulfuric acid, sodium hydroxide, hydrochloric acid, ethanol, petroleum ether, zinc acetate, concentrated nitric acid, 1,10-phenanthroline monohydrate, ferrous sulfate, Trichloroacetic acid (TCA), L-leucine (L-Leu), bovine serum albumin (BSA) and other reagents were purchased from China National Pharmaceutical Co., Ltd. (Beijing, China). 1,1-Diphenyl-2-picrylhydrazine (DPPH), mixed standard of 35 fatty acids was obtained from Sigma-Aldrich (St. Louis, MO, USA).

### 2.2. Preparation of Snakehead Fish Soup

We adopted a traditional method to prepare snakehead soup (according to the method of Zhang, et al. [15]), with some improvements. Fresh fish meat (about 700 g) with head and bones were washed with tap water, and then the water was drained off. Luhua peanut oil (3% of the weight of the fish) was added into an iron pot and heated by an induction cooker (RT2134, Midea group, Foshan, China). After boiling, the fish meat was added into to the pan and stir-fried until golden brown. Water (3 times the weight of fish) was poured into the pan. The mixture was heated up to boiling temperature under 2100 W power, and kept boiling for 15 min. Foam in the surface was removed during the process. After the pan was covered by a lid, the mixture was heated under mild power (300 W) for 30 min. Salt (1.2% of the weight of the fish) was added right before turning the induction cooker off. The fish soup was separated by filtering through single layer cheese cloth.

With reference to the method of Guo, et al. [16], snakehead fish soup was dried in a pilot-scale vacuum freeze dryer (102241, Martin Christ, Osterode, Germany). The dryer was equipped with 10 stainless steel pans (50 cm diameter). The wild and farmed snakehead soup were transferred to plates, cooled to room temperature, and then frozen at −18 °C. The frozen fish soup was dried under the conditions of shelf temperature at 20 °C and a vacuum of 338 Pa for 4 h, followed by drying at −5 °C for 6 h, to remove most of the water. After adjusting the temperature and vacuum to −1 °C and 338 Pa, respectively, the fish soup was further dried for 10 h. The dried snakehead fish soup was pulverized into powder, placed in a plastic bag, vacuum-packed, and stored in a desiccator at room temperature. The preparation of snakehead soup powder was repeated twice. The yield of soup powder made from farmed snakehead was about 14.97%, and that of wild snakehead 12.12%.

### 2.3. Proximate Composition

The moisture content of the samples was determined by drying the samples in an oven at 105 °C for 4–6 h [17]. Crude protein was determined by Kjeldahl method [17,18]. During the analysis, an automatic Kjeldahl analyzer (K9840, Haineng, Shandong, China) was used, and a factor of 6.25 was used to convert nitrogen content to protein content. Ash was determined by incinerating the sample in a muffle furnace (BF51794C-1, Thermo Scientific, Waltham, MA, USA) at 550 °C for 18 h [19]. Crude fat was extracted according to the method of Bukhanko, et al. [20].

The free amino acid content was determined with reference to the method of Bordons [21] with minor modifications. The lyophilized powders of wild and farmed snakehead soup were weighed and dissolved in distilled water to constant volume. The soup was put in a centrifuge tube. Two milliliters of 10% TCA solution was added to precipitate the protein for 10 min. Then, the sample was centrifuged at 4000 r/min for 15 min. Finally, the supernatant was diluted by 15 times. The absorbance was measured at 570 nm using a spectrophotometer (UV-1750, Shimadzu, Kyoto, Japan) using a ninhydrin colorimetric method.

The determination of TCA-soluble peptide content was performed with reference to the method of Rawdkuen, et al. [22]. About 0.5 g of fish broth freeze-dried powder was mixed in 18 mL of 5% TCA solution, homogenized at 11,000 r/min for 2 min, then kept at 4 °C for 1 h after homogenization, and then centrifuged at 8500 r/min for 5 min. The supernatant was assayed for TCA-soluble peptide concentration by Lowry’s method [23].

### 2.4. Total Amino Acid Composition Analysis

Referring to the method of Fuentes, et al. [5], the amino acid composition was determined by HPLC method. The sample was weighed into a hydrolysis tube for hydrolyzing. After this step, it was taken out and cooled. Then it was transferred to a colorimetric tube to constant volume and put it in a vacuum drying oven for drying. The centrifuge tube was filled with nitrogen, then reagents were added to derivatize at room temperature. Mobile phase was added and mixed well. The processed sample was passed through the filter membrane and entered the instrument (1525 + 717 + 2998PDA + RI, Waters, MA, USA) for analysis. The amino acid content was calculated according to the following formula
$$X=\frac{c\;\times \;V}{m}$$

X: the content of amino acids in the sample, in mg/ kg; m: the sample weight, in g; V: the constant volume of the sample after hydrolysis, in mL; c: the calculated concentration of each amino acid on the standard curve, in μg/ mL.

Amino acid score (AAS), chemical score (CS) and the Essential Amino Acid Index (EAAI) were calculated by the following equations.
$$\mathrm{AAS}=\frac{\mathrm{Sample}\;\mathrm{amino}\;\mathrm{acid}\;\mathrm{content}/\%}{\mathrm{Content}\;\mathrm{of}\;\mathrm{the}\;\mathrm{same}\;\mathrm{amino}\;\mathrm{acid}\;\mathrm{in}\;\mathrm{the}\;\mathrm{FAO}/\mathrm{WHO}\;\mathrm{scoring}\;\mathrm{standard}\;\mathrm{model}/\%}$$
$$\mathrm{CS}=\frac{\mathrm{Sample}\;\mathrm{amino}\;\mathrm{acid}\;\mathrm{content}/\%}{\mathrm{Identical}\;\mathrm{Amino}\;\mathrm{Acid}\;\mathrm{Content}\;\mathrm{in}\;\mathrm{Whole}\;\mathrm{Egg}\;\mathrm{Protein}/\%}$$
$$\mathrm{EAAI}=\sqrt[{n}]{\frac{100a}{\mathrm{ae}}\;\times \;\frac{100b}{\mathrm{be}}\;\times \;\cdots \;\times \;\frac{100j}{\mathrm{je}}}$$

Among them, n is the number of essential amino acids compared; a, b… j are the essential amino acid content of the protein /%; ae, be…je is the essential amino acid content of whole egg protein /%.

### 2.5. Fatty Acid Composition Analysis

Referring to the method of Zhang, et al. [24], the fatty acid composition was determined by GC-MS method. An appropriate amount of the sample was weighed. Pyrogallic acid (100 mg), 95% ethanol (2 mL) hydrochloric acid (10 mL), and zeolite were added to the flask with the sample. Then the flask was soaked in water at 70~80 °C for 40 min.

After hydrolysis, 95% ethanol (10 mL) was added to the sample, and 50 mL diethyl ether petroleum ether mixture and hydrolysate were merged into a separating funnel. After shaking and standing still, the ether layer extract was collected. The above steps were repeated 3 times. Finally, it was collected into a flask of constant weight. The flask was dried in a water bath and then dried at 100 °C ± 5 °C for 2 h.

The extracted fat was subjected to fat saponification and fatty acid methylation. The processed sample was passed through the filter membrane (0.45 μm) and entered the instrument (Trace1310 ISQ, Thermo Scientific, Waltham, MA, USA) for analysis.

Chromatographic column: TG-5MS (30 m × 0.25 mm × 0.25 μm); heating procedure: keep at 80 °C for 1min, then rise to 200 °C at the rate of 10 °C /min, continue to rise to 250 °C at the rate of 5 °C /min, and finally rise to 270 °C at the rate of 2 °C /min, and keep for 3 min; inlet temperature: 290 °C; carrier gas flow rate: 1.2 mL/min; mass spectrometry conditions: ion source temperature: 280 °C, transmission line temperature: 280 °C, solvent delay time: 5 min, scanning range: 30~400 amu, ion source: EI source 70 eV.

### 2.6. Mineral Element Composition Analysis

With reference to the method of Leme, et al. [25], the mineral composition was determined by the ICP-MS method. An appropriate amount of sample was weighed into the digestion tank and then mixed with nitric acid. After the reaction, it was put into a microwave digestion apparatus for digestion. After the temperature was cooled, the digestion tank was taken out and put in a fume hood. Then the digestion tank was opened, and it was diluted with ultrapure water to a certain concentration. The blank control was treated in the same way. Samples were tested on an ICP-MS instrument (iCAPQ, Thermo Scientific, Waltham, MA, USA).

### 2.7. Measurement of Antioxidant Ability

#### 2.7.1. DPPH Radical-Scavenging Ability

Measurement of DPPH radical-scavenging ability was carried out referring to the method of Zhang, et al. [24]. Fish soup powder was weighed and dissolved in distilled water and diluted. Then it was transferred into a centrifuge tube and centrifuged. A 4.0 mL supernatant was mixed with 1.0 mL DPPH solution (0.1 mmol/L in 95% ethanol). The mixture was shaken and left for 30 min at room temperature, and the absorbance of the resulting solution was measured at 517 nm using a spectrophotometer (UV-1750, Shimadzu, Kyoto, Japan). Distilled water was used as the control group instead of the sample, and ethanol was used as the blank group. The DPPH radical-scavenging ability was calculated according to the following formula.
DPPH Radical-Scavenging (%) = [(OD_1_ − (OD_2_ − OD_3_))/OD_1_] ∗ 100 

Among them, OD_1_, OD_2_, and OD_3_ represent the absorbance values of the control group, sample group and blank group, respectively.

#### 2.7.2. Fe^2+^ Chelating Ability

We referred to the method of Zhang, et al. [24] for the determination of Fe^2+^ chelating ability with some modifications. Fish soup powder sample was weighed and dissolved in distilled water and diluted. Then it was transferred into a centrifuge tube and centrifuged. The supernatant (1 mL) was mixed with 3.85 mL of distilled water, ferrous chloride (0.05 mL, 2 mmol/L) and ferrozine (0.1 mL, 5 mmol/L). After vigorous mixing, the solution was left to stand at room temperature for 20 min, and the absorbance of the resulting solution was measured at 562 nm with a spectrophotometer (UV-1750, Shimadzu, Kyoto, Japan). The samples and chemical reagents were replaced by distilled water as the control and blank groups, respectively. The Fe^2+^ chelating ability was calculated according to the following formula.
Fe^2+^ Chelating (%) = [(OD_1_ − (OD_2_ − OD_3_))/OD_1_] ∗ 100

Among them, OD_1_, OD_2_ and OD_3_ represent the absorbance values of the control group, the sample group, and the blank group, respectively.

#### 2.7.3. Hydroxyl Radical-Scavenging Ability

It was referred to the method of Zhang, et al. [24] with some modifications. Fish soup powder sample was weighed and dissolved in distilled water and diluted. Then it was transferred into a centrifuge tube and centrifuged. The supernatant (2 mL) mixed with 0.75 mM 1,10-phenanthroline (2 mL), 0.75 mM FeSO_4_ (2 mL), and 0.2 M phosphate buffer (pH = 7.4, 2 mL). Then 0.12% H_2_O_2_ (1 mL) solution was added to the mixture and incubated at 37 °C for 1 h, and the absorbance of the resulting solution was measured at 536 nm with a spectrophotometer (UV-1750, Shimadzu, Kyoto, Japan). Control group 1 consisted of the same solution as the sample group, except that equivalent deionized water was used instead of the sample solution; control group 2 was based on control group 1 by replacing H_2_O_2_ with distilled water. The composition of the blank group was the same as that of the sample group, except that equivalent deionized water was used instead of 1,10-phenanthroline monohydrate and FeSO_4_ solution. The hydroxyl radical-scavenging ability was calculated according to the following formula.
Hydroxyl Radical-Scavenging (%) = (OD_s_ − OD_b_ − OD_1_)/(OD_2_ − OD_1_) ∗ 100%

Among them, OD_s_, OD_b_, OD_1_ and OD_2_ represent the absorbance of the sample group, blank group, control group 1 and control group 2, respectively.

### 2.8. Statistical Analysis

The experimental data were expressed as mean ± standard deviation (mean ± SD). Multiple comparison analysis was performed using SPSS 26.0. Each index was repeated three times, and the minerals were repeated two times.

## 3. Results

### 3.1. Proximate Composition

It can be seen from Table 1 that the proximate composition of moisture, protein, fat, ash, free amino acids, and soluble peptides in the farmed snakehead fish soup (wet base) were 95.01%, 2.55%, 0.89%, 0.92%, 0.47%, and 0.62%, respectively. Zhu, et al. [26] reported that the proximate composition of protein, fat, ash in snakehead fish soup were 0.5–0.9%, 0.6%, and 1.4%, respectively. With respect to the soup prepared by Zhu, et al. [26], the ash content of the snakehead fish soup in this study (Table 1) was relatively lower, but the crude protein and crude fat content were relatively higher. Compared with the soup making method by Zhu, et al. [26], the heating power used in this study is stronger, and therefore the protein and fat were more easily dissolved from the fish meat into the soup. Zhu, et al. [26] chosen a fish to water mass ratio of 1:6 to make soup. The fish to water mass ratio (1:3) was smaller in our study (Method 2.2). The study of Xu [27] showed that when the fish-water mass ratio was 1:4, the protein concentration was the highest in the soup; when the ratio was 1:10, the concentration was the lowest. This was because a large amount of water diluted the concentration of dissolved protein. Therefore, the fish soup prepared in our study (Table 1) contained more protein and crude fat. Zhu, et al. [26] used the fish head as the raw material to make fish soup. We used the descaled and gutted snakehead fish as the raw material to make fish soup. The fish head contains higher ash content. As a result, the ash content of our soup was much lower.

In China, in addition to snakehead fish, crucian carp, bighead carp, etc. are also freshwater fish commonly used in making soups. According to the reports of Xia and Xu [28] and Xu [27], the protein, fat, and ash in the crucian carp (*Carassius auratus*) soup and bighead carp (*Aristichthys nobilis*) soup (wet base) were in the range of 0.70%~1.11%, 0.70%~1.11% and 1.2%~1.6%, respectively. By comparing with the basic nutrients of the fish soups mentioned above, it might be concluded that the snakehead fish soup possesses the characteristic of high protein and fat contents, but low content of ash.

The proximate composition of moisture, protein, fat, ash, free amino acids, and soluble peptides in the wild snakehead fish soup (wet base) were 95.96%, 1.90%, 0.79%, 0.71%, 0.30%, and 0.47%, respectively. Compared with the farmed snakehead soup, its crude fat, crude ash, free amino acid, and soluble peptide content were slightly lower (*p* > 0.05); but crude protein content was significantly lower (*p* < 0.05). The total amount of protein and its degradation products in the farmed snakehead soup was higher than that of wild snakehead soup. It might be related to the relatively higher protein content in the muscle of farmed snakehead [14]. Yin, et al. [29] and O’Neill, et al. [6] and Cahu, et al. [7] reported that the mass fractions of protein and fat in muscles of fish in different growth environments varied greatly. The nutritional content of fish is closely related to its living environment (farmed or wild growth), feed composition, and growth period (larvae or adults, etc.). The fat content of feed for farmed snakehead fish is higher than that in the food of wild snakehead fish (Appendix A). Increasing the fat level in the feed could increase the energy of farmed snakehead fish, which might reduce the energy supply proportion of protein and thus increase the protein content in fish meat [30]. As a result, the protein content of the soup made from the farmed snakehead fish was higher.

### 3.2. Total Amino Acid Composition

It can be seen from Table 2 that the amino acid of the highest content in the farmed snakehead soup (dry base) is glycine (82.56 mg/g, 21.16%), followed by arginine (42.79 mg/g, 10.97%), alanine (39.13 mg/g, 10.03%), proline (39.05 mg/g, 10.01%) and glutamic acid (36.20 mg/g, 9.28%), the lowest content was tyrosine (3.81 mg/g, 0.98%). Correspondingly, the content of glycine in farmed snakehead soup (wet base) was 4.12 mg/g, followed by arginine 2.13 mg/g, alanine 1.95 mg/g, proline 1.95 mg/g and glutamic acid 1.81 mg/ g, the lowest content was tyrosine 0.19 mg/g. Zhu, et al. [26] reported that the highest content of snakehead soup (wet base) was glycine (2.011 mg/g), which was consistent with our result.

The total amino acid content of the farmed snakehead soup (dry base) was 390.11 mg/g, and the corresponding total amino acid content in wet basis was 19.46 mg/g. The lysine content of the farmed snakehead soup (dry base) was 30.15 mg/g, the corresponding wet basis content was 1.50 mg/g, and the percentage was 7.73%. Zhu, et al. (2017) reported that the total amino acid content of snakehead fish soup was 10.177 mg/g, the lysine content was 0.635 mg/g, and the percentage was 6.24%. Compared with the snakehead fish soup reported by Zhu, et al. [26], the total amino acid content and lysine content of the farmed snakehead soup in our study (Table 2) were significantly higher, which were the same as the differences in free amino acids in the proximate composition (Table 1).

Zhang, et al. [31] reported that the total amino acid content of crucian carp soup was 0.66–0.70 mg/g. The essential amino acid content of the soup in this study (Table 2) was 107.63 mg/g, and the corresponding wet basis content was 5.37 mg/g, accounting for 27.59%. The essential amino acid content of crucian carp soup reported by Zhang, et al. [31] was 24.97% to 25.85%. The total amino acid and essential amino acid content of snakehead fish soup in this study were higher than those reported by Zhang, et al. [31] in crucian carp soup. Therefore, compared with crucian carp soup, snakehead fish soup’s nutritional value of amino acids was higher.

Compared with farmed snakehead fish soup, the content of serine, histidine, arginine, threonine, proline, methionine, isoleucine, leucine, and phenylalanine in wild snakehead soup was relatively lower (*p* > 0.05). The contents of aspartic acid and glutamic acid (*p* > 0.05) were relatively higher. The contents of glycine, alanine, tyrosine, valine and lysine (*p* < 0.05) were significantly lower. In terms of percentages, except for aspartic acid, glutamic acid, and arginine, there was no significant difference. The total amino acid content (dry basis) of wild snakehead fish soup was 358.97 mg/g, which was slightly lower than that of the farmed snakehead fish soup (*p* > 0.05). The result was consistent with the protein content (Table 1). The essential amino acid content of wild snakehead soup was 99.42 mg/g, the percentage was 27.70%. There was no significant difference between the wild and farmed snakehead fish soup (*p* > 0.05). Zhao, et al. [32] reported that there was no significant difference in the total amino acid content and composition of farmed and wild snakehead fish meat, which was consistent with the results of this study. The results of Table 2 showed that the growth environment had no significant effect on the total amino acid content and composition of snakehead fish soup.

It can be seen from Table 3 that the AAS and CS of the lysine of the two kinds of snakehead soup were greater than 1. Lysine is the first limiting amino acid of cereals, which is the staple food of Asian people [33]. Eating fish soup and grains at the same time can achieve nutritional balance. The AAS close to 1 were threonine, phenylalanine + tyrosine, and leucine. Those below 1 were valine, isoleucine, methionine + cysteine, and the lowest among them were valine and isoleucine, which were the first type of restrictive amino acids. Compared with the farmed snakehead soup, except the AAS and CS of methionine + cysteine were significantly higher (*p* < 0.05), the AAS and CS of essential amino acids in the wild snakehead soup had no significant difference (*p* > 0.05).

### 3.3. Fatty Acid Composition

The GC-MS chromatogram of fatty acid analysis is shown in Appendix A, and the calculated individual fatty acid content is in Table 4. It can be seen from Table 4 that a total of 25 kinds of fatty acids were detected in the farmed snakehead soup, including 20 kinds of saturated fatty acids (SFA), 5 kinds of monounsaturated acids (MUFA) and 8 kinds of polyunsaturated acids (PUFA). The highest content was C18:1n9c, which was 5.29 g/100 g. The relatively low content was C15:0, C17:0, C18:3n6, C21:0, C20:2, C20:3n6, C20:3n3, C22:1n9, C20:4n6, C23:0, C22:2, C20:5n3 and C24:1. Their content was in the range of 0.01~0.06 g/100 g. The content of EPA+DHA was 0.24 g/ 100 g, accounting for 1.77%. The total amount of MUFA was 5.78 g/100 g, accounting for 42.33%. The total amount of n-6 PUFA and n-3 PUFA were 3.50 g/100 g and 0.41 g/100 g, respectively, and the total proportion were 25.67% and 2.98%, respectively.

Twenty-four kinds of fatty acids were detected in wild snakehead soup. Compared with the fatty acids detected in farmed snakehead soup, the undetected fatty acid was C22:2. The contents of C18:0, C18:1n9c and C20:3n6 in wild snakehead soup were significantly higher than those in farmed snakehead soup (*p* < 0.05); while the contents of C20:3n3 and C24:1 were significantly lower than those in farmed snakehead soup (*p* < 0.05). There was no significant difference in other fatty acids content (*p* > 0.05). The total amount of fatty acids in wild snakehead soup was 16.22 g/100 g, which was 1.19 times that of farmed snakehead fish soup. The difference was the same as the fat content in dry basis (Table 1).

The absolute contents of EPA+DHA and ∑MUFA In wild snakehead soup were 0.39 g/100 g and 7.17 g/100 g, respectively. The absolute content of EPA+DHA was higher than that of farmed snakehead fish soup (*p* > 0.05), and the absolute content of MUFA was significantly higher than that of farmed snakehead fish soup (*p* < 0.05). Compared with farmed fish, wild fish usually contains higher levels of eicosapentaenoic acid (EPA) and docosahexaenoic acid (DHA). Several studies have pointed out that the content of MUFA in wild fish is higher [6]. The results of this study (Table 3) were consistent with the above results. According to the reported research differences in diet and growth environment lead to differences in the fatty acid composition of fish muscle between the wild and farmed fish [6,34,35]. Farmed snakehead fish swims on the surface layer of fish ponds with high temperature. However, the wild snakehead fish stay at the bottom layer of water with low temperature and high activity. Studies have shown that the greater the depth, the more total polyunsaturated fatty acids [36]. It was speculated that the different growth environment might change the fatty acid composition of snakehead fish muscles, thereby resulting in different fatty acid composition of the soups made from farmed and wild snakehead fish.

n-3 polyunsaturated fatty acids (n-3 PUFAs) have good anti-inflammatory properties [37]; some n-6 polyunsaturated fatty acids are required for normal human metabolism and human health, while n-6 polyunsaturated fatty acids have pro-inflammatory activity and play an important role in immune function [38]. Studies have shown that a lower n-6 PUFAs to n-3 PUFAs ratio is beneficial to human health [39,40]. The proportion of n-6 PUFAs to n-3 PUFAs in the diet of the population in developed countries is high (15:1-20:1) [40]. The ratio of n-6 polyunsaturated fatty acids and n-3 polyunsaturated fatty acids in the farmed fish snakehead soup in this study is 8.61, and 8.40 in wild snakehead fish soup. The results indicated that snakehead fish soup is a good source of polyunsaturated fatty acids.

### 3.4. Mineral Composition

Calcium is essential for the various basic physiological functions such as bone mineralization, blood coagulation, neuronal transmission, muscle contraction, and intracellular signaling [41]; Iron is an important part of human hemoglobin, myoglobin, which are involved in the transportation and storage of human oxygen and the synthesis of various metalloenzymes [42]. Zinc deficiency can cause metabolic disorders and decreased immune function, leading to bacterial, viral, and fungal infections, growth retardation, premature and poor wound healing [42]. It can be seen from Table 4 that the contents of the four trace elements Cu, Fe, Zn and Mn in the farmed snakehead soup were 0.55 mg/kg, 13.00 mg/kg, 9.04 mg/kg and 0.66 mg/kg, respectively, and the content of the major element Ca was 1.13 mg/g. Zhang, et al. [24] reported that the content of Ca, Zn, and Fe in snakehead fish soup were 0.81 mg/kg, 0.25 mg/kg and 0.15 mg/kg, respectively, which were lower than the mineral contents detected in this study. This might be due to the higher heating power during the processing of snakehead soup in this study, which was beneficial to the dissolution of minerals. At the same time, the choice of pot and boiling water may have a great impact on the mineral content. Tang, et al. [43] reported that the Ca, Fe and Zn contents in crucian carp soup were 0.08 mg/g, 1.41 mg/kg and 4.81 mg/kg, respectively. Through the comparative analysis with the above experimental results, it can be seen that the mineral content of the snakehead fish soup was high. Experience and experiments have proved that the snakehead fish soup has the effects of promoting wound healing [12,13]. Some scholars infer that the high Zn content in snakehead soup is one of the main factors for its outstanding wound-healing effect [13].

Compared with farmed snakehead soup, the contents of Cu, Fe, Mn, and Ca in wild snakehead soup were 0.43 mg/kg, 11.00 mg/kg, 0.32 mg/kg and 0.83 mg/g respectively, which were all significantly lower (*p* < 0.05). However, the Zn content was significantly higher (*p* < 0.05). O’Neill, et al. [6] showed that compared with farmed yellowtail, the Ca content of wild yellowtail was significantly lower (*p* < 0.05) but the Zn content was significantly higher (*p* < 0.05). Their findings were consistent with this study (Table 5).

### 3.5. Antioxidant Ability

The wound repair process can induce cellular oxidative stress and generate various free radicals. In turn, the oxidative stress may seriously interfere with wound healing through skin damage, neuropathy, and local infection [44,45]. The anti-oxidative ability of snakehead soup mainly comes from antioxidant peptides [46], which can reduce oxidative stress by scavenging reactive oxygen species and chelating transition metals, thereby accelerating wound healing [47].

The antioxidant ability of soup made from farmed and wild snakehead fish was shown in Figure 1. The DPPH radical-scavenging ability of the farmed snakehead soup was 57.89%, the hydroxyl radical-scavenging ability was 25.61%, and the Fe^2+^ chelating ability was 21.21%. Zhang, et al. [24] reported that the DPPH radical-scavenging ability of snakehead soup was 75%, the hydroxyl radical-scavenging ability was 26%, and the Fe^2+^ chelating ability was 26%. The research results of Zhang, et al. [24] showed that the DPPH radical-scavenging ability was 1.3 times that of farmed snakehead soup in this study, and the hydroxyl radical scavenging ability and Fe^2+^ chelating ability were slightly lower. The lower antioxidant capacity of the fish soup in this study might be due to the fact that the heating power of the fish soup in this subject was higher, so the biologically active substances in the fish soup might be decomposed, resulting in lower antioxidant capacity [48].

The DPPH radical-scavenging ability of wild snakehead soup was 56.90%, the hydroxyl radical-scavenging ability was 22.54%, and the Fe^2+^ chelating ability was 23.36%. There was no significant difference in Fe^2+^ chelation ability and DPPH radical-scavenging ability of wild snakehead soup (*p* > 0.05), while the hydroxyl radical-scavenging ability was slightly lower (*p* < 0.05). The results showed that there was no obvious difference in the antioxidant capacity of the two soups.

## 4. Conclusions

Snakehead fish soup was rich in protein, lysine, and trace element Zn, and has high antioxidant activity. There was no obvious difference in the nutritional composition and antioxidant activity of soup made from snakehead fish in different growth environments. However, the crude protein content of farmed snakehead soup was higher, while the total fatty acid, MUFA and Zn content of wild snakehead soup were significantly higher, which may be related to the differences in food and environment. Further research is needed on the effect of growth environment on the health effects of snakehead fish soup, especially the functionality of wound healing.

## Figures and Tables

**Figure 1 foods-11-03294-f001:**
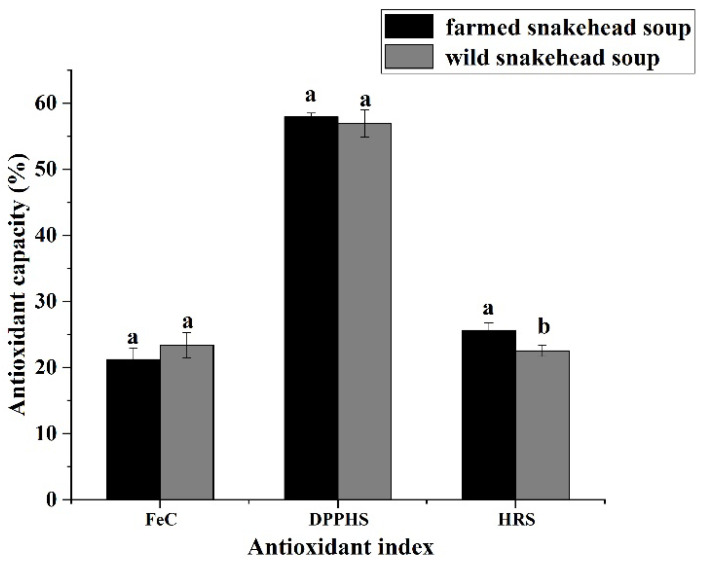
Antioxidant ability of soup made from farmed and wild snakehead fish. FeC: Fe2+ chelating ability, DPPHS: DPPH scavenging ability, HRS: hydroxyl radical scavenging ability. Different lowercase letters indicate significant difference in antioxidant capacity between the soup made from farmed and wild snakehead fish (*p* < 0.05).

**Table 1 foods-11-03294-t001:** Proximate composition of soup made from farmed and wild snakehead fish.

Composition	Farmed Snakehead Fish Soup		Wild Snakehead Fish Soup	
	Wet Base	Dry Base	Wet Base	Dry Base
Moisture	95.01 ± 0.32 ^a^	-	95.96 ± 0.90 ^a^	-
Crude fat	0.89 ± 0.09 ^a^	17.86 ± 1.82 ^a^	0.79 ± 0.03 ^a^	19.65 ± 2.42 ^a^
Crude protein	2.55 ± 0.08 ^a^	51.20 ± 1.62 ^a^	1.90 ± 0.01 ^b^	47.03 ± 0.36 ^a^
Crude ash	0.92 ± 0.12 ^a^	18.41 ± 2.47 ^a^	0.71 ± 0.11 ^a^	17.66 ± 2.80 ^a^
Free amino acids	0.47 ± 0.05 ^a^	9.49 ± 0.99 ^a^	0.30 ± 0.03 ^b^	7.30 ± 0.74 ^a^
Soluble peptide	0.62 ± 0.04 ^a^	12.47 ± 0.74 ^a^	0.47 ± 0.08 ^a^	11.69 ± 1.93 ^a^

Note: Proximate composition is expressed as %. Mean ± SD (standard deviation) from three replicates. Different lowercase letters in the same line indicate significant differences in composition between soup made from farmed and wild snakehead fish (*p* < 0.05).

**Table 2 foods-11-03294-t002:** Amino acid composition of soup made from farmed and wild snakehead fish.

Amino Acids	Farmed Snakehead Fish Soup		Wild Snakehead Fish Soup	
	mg/g	%	mg/g	%
Aspartic acid	17.20 ± 6.34 ^a^	4.41	19.50 ± 5.49 ^a^	5.43
Glutamic acid	36.20 ± 10.03 ^a^	9.28	36.80 ± 8.06 ^a^	10.25
Serine	14.25 ± 1.68 ^a^	3.65	13.61 ± 1.18 ^a^	3.79
Glycine	82.56 ± 1.53 ^a^	21.16	76.41 ± 1.44 ^b^	21.29
Histidine	7.48 ± 0.97 ^a^	1.92	6.78 ± 0.86 ^a^	1.89
Arginine	42.79 ± 7.10 ^a^	10.97	33.77 ± 2.66 ^a^	9.41
Threonine *	14.59 ± 0.96 ^a^	3.74	13.93 ± 0.56 ^a^	3.88
Alanine	39.13 ± 0.45 ^a^	10.03	34.74 ± 0.21 ^b^	9.68
Proline	39.05 ± 3.04 ^a^	10.01	34.30 ± 3.01 ^a^	9.55
Tyrosine *	3.81 ± 0.02 ^a^	0.98	3.64 ± 0.05 ^b^	1.01
Valine *	10.98 ± 0.15 ^a^	2.81	10.07 ± 0.14 ^b^	2.80
Methionine *	7.92 ± 0.32 ^a^	2.03	7.58 ± 0.30 ^a^	2.11
Cysteine	-	-	-	-
Isoleucine *	8.68 ± 0.40 ^a^	2.22	7.98 ± 0.11 ^a^	2.22
Leucine *	19.36 ± 0.74 ^a^	4.96	18.28 ± 0.09 ^a^	5.09
Phenylalanine *	15.95 ± 0.82 ^a^	4.09	14.35 ± 0.40 ^a^	4.00
Lysine *	30.15 ± 0.97 ^a^	7.73	27.23 ± 0.60 ^b^	7.59
TAA	390.11 ± 10.38 ^a^	100%	358.97 ± 15.92 ^a^	100%
EAA	107.63 ± 3.71 ^a^	27.59%	99.42 ± 0.32 ^a^	27.70%
NEAA	282.48 ± 14.08 ^a^	72.41%	259.55 ± 15.92 ^a^	72.30%
E/N		38.10%		38.31%

Mean ± SD (standard deviation) from two replicates. * indicates essential amino acids. “-” indicates that the amino acid is not detected. TAA indicates total amino acids, EAA essential amino acids, NEAA non-essential amino acids, E/N the ratio of essential amino acids to non-essential amino acids. Different lowercase letters in the same line indicate significant differences in the amino acid content between soup made from farmed and wild snakehead fish (*p* < 0.05).

**Table 3 foods-11-03294-t003:** Scoring AAS, CS and EAAI of protein in soup made from farmed and wild snakehead fish.

Amino Acid	FAO/WHO Amino Acid Scoring Standard Model/%	Whole Egg Protein Scoring Mode/%	Farmed Snakehead Fish Soup		Wild Snakehead Fish Soup	
			AAS	CS	AAS	CS
Threonine	4.00	4.98	0.94 ± 0.09 ^a^	0.75 ± 0.07 ^a^	0.97 ± 0.08 ^a^	0.78 ± 0.06 ^a^
Valine	5.00	7.42	0.56 ± 0.02 ^a^	0.38 ± 0.01 ^a^	0.56 ± 0.02 ^a^	0.38 ± 0.01 ^a^
Isoleucine	4.00	6.60	0.56 ± 0.04 ^a^	0.34 ± 0.02 ^a^	0.56 ± 0.03 ^a^	0.34 ± 0.02 ^a^
Leucine	7.00	8.80	0.71 ± 0.04 ^a^	0.56 ± 0.04 ^a^	0.73 ± 0.03 ^a^	0.58 ± 0.02 ^a^
Lysine	5.50	6.40	1.41 ± 0.08 ^a^	1.21 ± 0.07 ^a^	1.38 ± 0.04 ^a^	1.19 ± 0.03 ^a^
Methionine + Cysteine	3.50	5.48	0.58 ± 0.01 ^b^	0.37 ± 0.01 ^b^	0.60 ± 0.00 ^a^	0.39 ± 0.00 ^a^
Phenylalanine + Tyrosine	6.00	10.08	0.85 ± 0.06 ^a^	0.50 ± 0.03 ^a^	0.84 ± 0.05 ^a^	0.50 ± 0.03 ^a^
Tryptophan	1.00	1.70	-	-	-	-
Total	36.00	51.46				
EAAI			53.25			53.76

Note: AAS indicates amino acid scores, CS chemical scores, EAAI essential amino acid indices. “-” indicates the amino acid was not detected. Different lowercase letters in the same line indicate significant differences in amino acid scores and chemical scores between soup made from farmed and wild snakehead fish (*p* < 0.05).

**Table 4 foods-11-03294-t004:** Fatty acid composition in soup made from farmed and wild snakehead fish.

Fatty Acids	Farmed Snakehead Fish Soup		Wild Snakehead Fish Soup	
	g/100 g	%	g/100 g	%
C8:0	-	-	-	-
C10:0	-	-	-	-
C11:0	-	-	-	-
C12:0	-	-	-	-
C13:0	-	-	-	-
C14:0	0.12 ± 0.02 ^a^	0.85	0.11 ± 0.09 ^a^	0.65
C14:1	-	-	-	-
C15:0	0.02 ± 0.00 ^a^	0.13	0.02 ± 0.01 ^a^	0.11
C15:1	-	-	-	-
C16:0	2.46 ± 0.13 ^a^	18.00	2.88 ± 0.38 ^a^	17.75
C16:1	0.23 ± 0.03 ^a^	1.66	0.22 ± 0.08 ^a^	1.38
C17:0	0.03 ± 0.01 ^a^	0.23	0.03 ± 0.01 ^a^	0.18
C17:1	-	-	-	-
C18:0	0.76 ± 0.01 ^b^	5.58	1.00 ± 0.05 ^a^	6.16
C18:1n9t	-	-	-	-
C18:1n9c	5.29 ± 0.15 ^b^	38.77	6.44 ± 0.10 ^a^	39.69
C18:2n6t	-	-	-	-
C18:2n6c	3.41 ± 0.18 ^a^	24.98	3.67 ± 0.28 ^a^	22.64
C20:0	0.16 ± 0.02 ^a^	1.16	0.19 ± 0.00 ^a^	1.17
C18:3n6	0.03 ± 0.00 ^a^	0.20	0.03 ± 0.01 ^a^	0.17
C18:3n3	0.15 ± 0.00 ^a^	1.07	0.07 ± 0.03 ^a^	0.42
C20:1	0.20 ± 0.02 ^a^	1.49	0.19 ± 0.04 ^a^	1.15
C21:0	0.01 ± 0.00 ^a^	0.04	0.01 ± 0.00 ^a^	0.03
C20:2	0.06 ± 0.00 ^a^	0.44	0.05 ± 0.01 ^a^	0.29
C22:0	0.23 ± 0.06 ^a^	1.69	0.30 ± 0.03 ^a^	1.83
C20:3n6	0.02 ± 0.00 ^b^	0.17	0.04 ± 0.00 ^a^	0.24
C20:3n3	0.019 ± 0.000 ^a^	0.14	0.003 ± 0.000 ^b^	0.02
C22:1n9	0.04 ± 0.01 ^a^	0.32	0.31 ± 0.18 ^a^	1.93
C20:4n6	0.04 ± 0.02 ^a^	0.32	0.13 ± 0.15 ^a^	0.80
C23:0	0.01 ± 0.00 ^a^	0.04	0.01 ± 0.00 ^a^	0.05
C22:2	0.01 ± 0.00 ^a^	0.07	-	-
C20:5n3	0.04 ± 0.01 ^a^	0.27	0.08 ± 0.10 ^a^	0.47
C24:0	0.11 ± 0.03 ^a^	0.80	0.14 ± 0.01 ^a^	0.87
C24:1	0.01 ± 0.00 ^a^	0.09	0.01 ± 0.00 ^b^	0.05
C22:6n3	0.21 ± 0.05 ^a^	1.50	0.31 ± 0.39 ^a^	1.93
Total fatty acid	13.64 ± 0.04 ^b^		16.22 ± 0.73 ^a^	
EPA+DHA	0.24 ± 0.06 ^a^	1.77	0.39 ± 0.49 ^a^	2.39
∑SATD	3.89 ± 0.01 ^a^	28.52	4.67 ± 0.49 ^a^	28.82
∑MUFA	5.78 ± 0.14 ^b^	42.33	7.17 ± 0.16 ^a^	44.20
∑PUFAω6	3.50 ± 0.16 ^a^	25.67	3.87 ± 0.13 ^a^	23.85
∑PUFAω3	0.41 ± 0.05 ^a^	2.98	0.46 ± 0.52 ^a^	2.84

Mean ± SD (standard deviation) from three replicates. EPA indicates eicosapntemacnioc acid, DHA docosahexaenoic acid, SATD saturated fatty acid, MUFA monounsaturated fatty acid, PUFAω6 omega 6 polyunsaturated fatty acids, PUFAω3 omega 3 polyunsaturated fatty acids. “-” indicates the fatty acid was not detected. Different lowercase letters in the same line indicate significant differences in the fatty acid content soup made from farmed and wild snakehead fish (*p* < 0.05).

**Table 5 foods-11-03294-t005:** Mineral composition of soup made from farmed and wild snakehead fish.

Minerals	Farmed Snakehead Fish Soup	Wild Snakehead Fish Soup
Cu * (mg/kg)	0.55 ± 0.00 ^a^	0.43 ± 0.02 ^b^
Fe * (mg/kg)	13.00 ± 0.08 ^a^	11.00 ± 0.57 ^b^
Zn *(mg/kg)	9.04 ± 0.14 ^b^	12.57 ± 0.53 ^a^
Mn *(mg/kg)	0.66 ± 0.00 ^a^	0.32 ± 0.02 ^b^
Ca (mg/g)	1.13 ± 0.03 ^a^	0.83 ± 0.02 ^b^

* indicates trace elements. Mean ± SD (standard deviation) from two replicates. Different lowercase letters in the same line indicate significant differences in the mineral content between soup made from farmed and wild snakehead fish (*p* < 0.05).

## Data Availability

Not applicable.

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
