# Peer review of "Comparative Study of the Nutritional Composition and Antioxidant Ability of Soups Made from Wild and Farmed Snakehead Fish (Channa Argus)"

_foods, 2022, doi:10.3390/foods11203294_

Round 1

Reviewer 1 Report

The manuscript “Comparative study of the nutritional composition and antioxidant ability of the soups made from wild and farmed snake-3 head fish (Channa argus)” from Ren et al. is an interesting and complete work with numerous references to other studies, which is useful for comparison purposes.

Some details that could be considered are as follows:

1.- After the preparation of soups, authors fully dry under vacuum and low temperature the liquid phase, but no data are provided about the weight of dry powder in comparison with the mass of fish used for such preparation. This is interesting from a point of view of yield.

2.- Regarding the elemental analysis, in particular the contribution of Calcium present in the cooking water was considered for the determination of its total concentration?

3.- The foam formed during cooking, which was removed (line 93) could be a source of nutrients not considered in further analyses?

4.- Please, change “machine” (line 163) by “instrument” or “system”. In addition, change “thermo” to “Thermo Scientific”

5.- “2+” in Fe2+ must be revised throughout the manuscript as superscript.

6.- Regarding the different fish/water mass ratio used by authors and other studies, it could be expected that higher ratio of water could have a higher extraction power. When using 1:10, lower concentration of protein was reported, but if full drying of soup is performed, finally the absolute content should be higher in the dry powder.

7.- Due to higher cooking power applied, lower antioxidative ability is obtained in comparison to previous studies. In order to have a real, significant effect in persons, are the portions reasonable or (for instance) 10 litres of soup should be consumed every day?

Author Response

1.- After the preparation of soups, authors fully dry under vacuum and low temperature the liquid phase, but no data are provided about the weight of dry powder in comparison with the mass of fish used for such preparation. This is interesting from a point of view of yield.

Authors: Thanks for your suggestion. The yield of farmed snakehead is about 14.97%, and that of wild snakehead is about 12.12%. They were added in the revised manuscript (Line 108-110).

2.- Regarding the elemental analysis, in particular the contribution of Calcium present in the cooking water was considered for the determination of its total concentration?

Authors: In this study, we adopted the traditional Chinese cooking techniques. The water used in the soup preparation was tap water. The calcium in the tap water is 0.01-0.03 mg/g. The calcium in the soups made from wild and farmed snakehead fish were 0.83 and 1.13 mg/g, respectively. Therefore the influence of calcium in the tap water can be ignored.

3.- The foam formed during cooking, which was removed (line 93) could be a source of nutrients not considered in further analyses?

Authors: In the traditional Chinese cooking, the foam in the soup will be removed, which not only avoids the impact on the quality of the soup, but also makes the soup more uniform. The remove of the form may influence the nutrients of the soup.

4.- Please, change “machine” (line 163) by “instrument” or “system”. In addition, change “thermo” to “Thermo Scientific”

Authors: We have replaced "machine" with "instrument" in line 165-166 of the text.

5.- “2+” in Fe2+ must be revised throughout the manuscript as superscript.

Authors: We have changed "2+" to superscript in the text.

6.- Regarding the different fish/water mass ratio used by authors and other studies, it could be expected that higher ratio of water could have a higher extraction power. When using 1:10, lower concentration of protein was reported, but if full drying of soup is performed, finally the absolute content should be higher in the dry powder.

Authors: We fully agree with the reviewer’s point on the issue of fish/water mass ratio. In other studies mentioned in this paper, the contents of nutrients in fish soups with different fish-water mass ratios were all wet base contents, contents of nutrients in dry base of fish soups with different fish-water mass ratios were not reported. Therefore, we compared contents of nutrients in wet base with the reported values.

7.- Due to higher cooking power applied, lower antioxidative ability is obtained in comparison to previous studies. In order to have a real, significant effect in persons, are the portions reasonable or (for instance) 10 litres of soup should be consumed every day?

Authors: No efficacy evaluation is made in this paper. At present, we are studying the effects of different concentrations of soup on cells and rats.

Reviewer 2 Report

The manuscript “Comparative study of the nutritional composition and antioxidant ability of the soups made from wild and farmed snakehead fish (Channa argus)” describes the production and characterization of fish soup from a snakehead fish. The authors analyzed a variety of nutritional and biofunctional parameters such as proximate composition, amino acid contents, fatty acid contents, minerals and antioxidants activity of fish soup. The work is interesting and the research design is sound enough. Moreover, the manuscript is well described and balanced. The field of the study matches with the journal and the information will be supportive for the related researcher and encourage manufacturing aquafood product. However, there are some mistakes and corrections are necessary .

Comments:

1.      Introduction: Page 1, Line: 28. The scientific name “Channa argus” should be Italic.

2.      Materials and reagents: Line 70-74, the morphological description is irrelevant in this section. Rather, this can be placed in introduction section.

3.      Line 87; It is needed to state the amount of raw fish taken for the study.

4.      Line 88; please mention the name of the oil.

5.      Line 112; Please add the specification of muffle furnace used for ash determination.

6.      Line 152; What was the method used for fat extraction? Write more detail please.

7.      Line 323; Fatty acid composition: Could you please add the GC-MS chromatogram of fatty acid analysis.

8.      Fig 1. The x-axis and Y-axis title should be added. Could you please compare with the value of standard, otherwise it is difficult to understand the free radical scavenging ability.

Author Response

  1. Introduction: Page 1, Line: 28. The scientific name “Channa argus” should be Italic.

Authors: Thanks for your suggestion. We've changed the font of "word" to italic.

  1. Materials and reagents: Line 70-74, the morphological description is irrelevant in this section. Rather, this can be placed in introduction section.

Authors: The morphological description was removed.

  1. Line 87; It is needed to state the amount of raw fish taken for the study.

Authors: Approximate 15 kg raw fish each was taken for the study. (Line 72).

  1. Line 88; please mention the name of the oil.

Authors: We have added the brand name of the oil.

  1. Line 112; Please add the specification of muffle furnace used for ash determination.

Authors: We used the thermo scientific BF51794C-1 muffle furnace and have added the corresponding information to the text.

  1. Line 152; What was the method used for fat extraction? Write more detail please.

Authors: We have added the details of the fat extraction.

  1. Line 323; Fatty acid composition: Could you please add the GC-MS chromatogram of fatty acid analysis.

Authors: We have added the chromatogram of fatty acids (Figure S2) in the supplement.

  1. Fig 1. The x-axis and Y-axis title should be added. Could you please compare with the value of standard, otherwise it is difficult to understand the free radical scavenging ability

Authors: The antioxidant capacity data of farmed and wild snakehead soup obtained in this paper was calculated by comparing with the blank group and the control group, and the settings of the blank group and the control group were determined by referring to other literature.